# Pediatric Clinical Classification System for use in Canadian inpatient settings

**Peter J. Gill**[1,2,3,4]*, **Thaksha Thavam**[3], **Mohammed Rashidul Anwar**[3], **Jingqin Zhu**[3], **Teresa To**[1,3,4], **Sanjay Mahant**[1,2,3,4], **on behalf of the Canadian Paediatric Inpatient Research Network (PIRN)**[¶]

1 The Hospital for Sick Children, Toronto, Ontario, Canada, 2 Department of Pediatrics, University of Toronto, Toronto, Ontario, Canada, 3 Child Health Evaluative Sciences, SickKids Research Institute, Toronto, Ontario, Canada, 4 Institute of Health Policy, Management and Evaluation, Dalla Lana School of Public Health, The University of Toronto, Toronto, Ontario, Canada

¶ Membership of the Canadian Paediatric Inpatient Research Network (PIRN) is provided in the Acknowledgments.
* peter.gill@sickkids.ca

**Data Availability Statement:** The dataset for this study is held securely in coded form at ICES. The ICES is a prescribed entity under the Ontario's Personal Health Information Privacy Act (PHIPA). As a prescribed entity, ICES is allowed to collect

## Abstract

### Background

A classification system that categorizes International Statistical Classification of Diseases and Related Health Problems, Tenth Revision (ICD-10) diagnosis codes into clinically meaningful categories is important for pediatric clinical and health services research using administrative data. While a Pediatric Clinical Classification System (PECCS) is available for the United States ICD-10 system (i.e, ICD-10-CM), differences in the ICD-10 system between countries limits PECCS use in Canada.

### Objective

To translate PECCS from ICD-10-CM to ICD-10-CA for use in Canada (PECCS-CA), and examine the utility of PECCS-CA in administrative data of pediatric hospital encounters in Ontario, Canada.

### Methods

PECCS was translated by mapping each ICD-10-CA code to its corresponding ICD-10-CM code, based on code description and alphanumeric code, using automated functions in Microsoft Excel. All unmatched ICD-10-CA codes were manually matched to an ICD-10-CM code. The ICD-10-CA codes were mapped to a PECCS category based on the placement of the corresponding ICD-10-CM code. Finally, in this cross-sectional study, the utility of PECCS-CA was examined in pediatric hospital encounters in children <18 years of age with an inpatient or same day surgery encounter, between April 1, 2014 to March 31, 2019 in Ontario.

### Results

In total, 16,992 ICD-10-CA diagnosis codes were mapped to 781 mutually exclusive condition categories that included pediatric specific conditions and treatments in PECCS-CA.

personally identifiable information for analysis that evaluate, plan, and/or monitor the health care system or for analysis related to the health or safety of the public without receiving individual consent. While legal data sharing agreements between ICES and data providers (e.g., healthcare organizations and government) prohibit ICES from making the dataset publicly available, access may be granted to those who meet pre-specified criteria for confidential access, available at www.ices.on.ca/DAS (email: das@ices.on.ca). Details on submitting a request form to access the data is found in the following hyperlink (https://www.ices.on.ca/DAS/Submitting-your-request). The full dataset creation plan is available from the authors upon request.

**Funding:** This study was funded through the New Investigator grant (21-01) from the Physicians' Services Incorporated (PSI) Foundation. The authors who received the award were PJG, TT, and SM. The funders had no role in study design, data collection and analysis, decision to publish, or preparation of the manuscript.

**Competing interests:** We have read the journal's policy and the authors of this manuscript have the following competing interests: PJG and SM have received grants from the Physicians' Services Incorporated (PSI) Foundation during the conduct of the study and grants from the Canadian Institute of Health Research (CIHR). PJG has also received personal fees from CIHR and EBMLive outside the submitted work. SM has also received personal fees from the Journal of Hospital Medicine outside the submitted work. The above indicated items does not alter our adherence to PLOS ONE policies on sharing data and materials.

From the 781 categories, 777 (99.5%) were derived from the original PECCS, 3 (0.4%) from merging the original PECCS categories, and 1 (0.1%) was newly developed. The PECCS-CA was applied to health administrative data of 911,732 hospital encounters in children. The most prevalent condition in children was low birth weight (n = 54,100 encounters).

## Conclusion

The PECCS-CA is an open-source classification system which maps ICD-10-CA codes into 781 clinically important pediatric categories. The PECCS-CA can be used for pediatric health services and outcomes research in Canada.

## Introduction

The large volume and cost of hospitalizations in Canadian children [1, 2] highlights the need to study this population to improve care and outcomes. In 2019, the Canadian Institute for Health Information (CIHI) reported that the provincial/territorial government hospital expenditure in Canada was over $64.2 billion dollars, with the hospital expenditure in children 19 years of age and younger to be over $6.8 billion [1]. These costs stem from over 260,000 inpatient hospitalizations observed in children in Canada [2]. Health administrative data is valuable for understanding the epidemiology of hospital use, and the reasons for admissions. The thousands of specific *International Statistical Classification of Diseases and Related Health Problems*, *Tenth Revision*, *Canada* (ICD-10-CA) diagnosis codes present, make it difficult to meaningfully analyze administrative data without using classification systems that map the specific codes into clinically relevant categories (e.g. pneumonia, depression). By grouping the diagnosis codes, researchers, payers, or policy makers can examine patterns in healthcare utilization and costs, develop patient cohorts for research, and answer important research questions using advanced observational study designs.

There are a few existing groupers that categorize ICD-10-CA diagnosis codes into clinical categories. One of these grouping methodologies is a translation of the Clinical Classifications Software (CCS) from the United States (US) ICD-10 codes (i.e. ICD-10-CM [Clinical Modification]) to the Canadian codes [3]. In this grouping methodology, the ICD-10-CA codes were mapped to 130 clinical categories of chronic health conditions [3]. Other grouping methodologies include the ICD-10-CA chapters which classifies diseases and related health problems and contains 23 broad category chapters [4], and the CIHI Case Mix Group (CMG) which categorizes acute care inpatients into clinically relevant groups using diagnosis and intervention codes from the patient's hospital record [5, 6]. However, limitations in these grouping methodologies such as only categorizing diagnosis codes into chronic health condition categories, or lacking important pediatric conditions prevent its' use in pediatric health services research. To date, a classification system that categorizes ICD-10-CA diagnosis codes in health administrative data into pediatric specific, mutually exclusive categories does not exist.

The Pediatric Clinical Classification System (PECCS) that categorizes the US ICD-10-CM codes into clinically distinctive categories currently exists [7]. The PECCS classifies 73,374 ICD-10-CM discharge diagnosis codes into 834 clinically distinctive categories, and identifies several important pediatric conditions (e.g. bronchiolitis, redundant prepuce and phimosis) including treatments (e.g. chemotherapy) [7, 8] which are missing from other classification systems. The PECCS was developed using the Healthcare Cost and Utilization Project (HCUP) Clinical Classifications Software (CCS) for ICD-10-CM diagnosis codes [9, 10] and

Keren *et al.*'s ICD-9-CM pediatric diagnosis code grouper [11]. In the US, PECCS has been used in studies to group diagnosis codes of pediatric hospital encounters into mutually exclusive condition categories in children's hospitals exclusively [12], and in general and children's hospitals [13] to identify high priority conditions based on prevalence and costs. It has also been used to identify high priority conditions in pediatric ambulatory surgeries [14], and to classify the comorbidities present in pediatric patients hospitalized with catatonia [15]. Several countries have created their own clinically modified ICD-10 classification system to address their country-specific needs [16]. For instance, the US ICD-10 system contains over 70,000 ICD-10-CM codes, while the Canadian system contains over 16,000 ICD-10-CA codes [16]. These country-specific modifications limits the use of classification systems such as PECCS or HCUP CCS outside of the US.

Therefore, the objective of this study was to translate PECCS from ICD-10-CM to ICD-10-CA for use in pediatric health services research in Canada (PECCS-CA). Additionally, we examined the use of PECCS-CA on health administrative data of pediatric hospital encounters in Canada's most populous province, Ontario.

## Methods

### Translation process of PECCS from ICD-10-CM to ICD-10-CA

The details behind the methodology used to develop the original PECCS for ICD-10-CM codes can be found in an existing research letter [7], and is also briefly presented in Fig 1. To translate PECCS, we first mapped each ICD-10-CA code to its corresponding ICD-10-CM code using automated functions in Microsoft® Excel® for Microsoft 365 MSO (Version 2111). Codes were first matched based on their code description. ICD-10-CA codes with the exact same or nearest match in code description to the ICD-10-CM codes were mapped

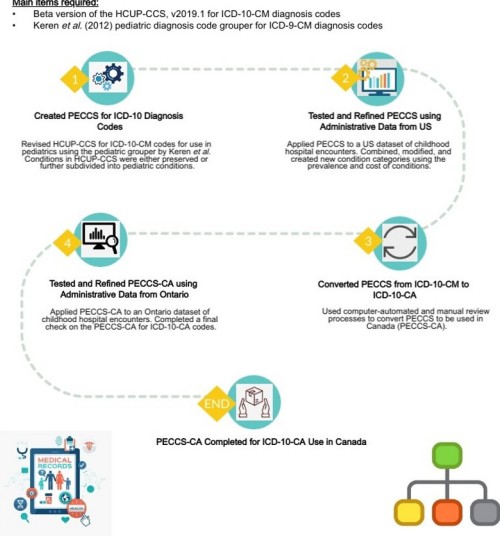

**Fig 1. Overview of the steps used to develop the original PECCS, the translation process to develop PECCS-CA, and its' application on administrative data of hospital encounters.** This figure presents an overview of the steps our research team used to develop PECCS, translate PECCS from ICD-10-CM to ICD-10-CA to be used in Canada (PECCS-CA), and its' application on administrative data of hospital encounters in Ontario. The references for the clip art pictures are presented below: 1) Clipart Library. (n.d.). Source: https://tinyurl.com/y69r2vm7; 2) PinClipart. (2018). Source: https://tinyurl.com/yxmdrv65; 3) Shutterstock. (2020). Source: https://tinyurl.com/y6a6qd29; 4) Tom Hand. (2019). Source: https://tinyurl.com/yyr2495o; 5) Convert Png To Icon. (2019). Source: https://tinyurl.com/y58snznv; 6) Pngitem. (2019). Source: https://tinyurl.com/y4hq8wa7; 7) SVG Repo. (n.d.). Source: https://tinyurl.com/yyvn6ons.

together. We also used the full alphanumeric code to match the ICD-10-CA codes exactly to ICD-10-CM based on face validity. Next, ICD-10-CA codes that did not match through the initial step were matched to the nearest 3- or 4-character ICD-10-CM code using Microsoft Excel. Finally, all remaining unmatched ICD-10-CA codes were manually mapped to ICD-10-CM codes by reviewing their code descriptions and ensuring that codes were congruent based on clinical judgement. Each ICD-10-CA code was mapped to a PECCS category based on where their corresponding ICD-10-CM code was placed. All ICD-10-CA codes were manually reviewed initially by one author (M.R.A), and then by three others (P.J.G, S.M, T.T), to either retain, merge, or create new categories ([Fig 2]). Any discrepancies in the translation process were resolved by consensus and discussed over meetings.

## Study design and data source

In this cross-sectional study, the use of PECCS-CA was examined by applying it on health administrative data of hospital encounters in children from all pediatric and general hospitals in Ontario. The data were obtained from linked health administrative databases housed at ICES. Datasets at ICES are linked using unique encoded identifiers known as the confidential ICES Key Number (IKN). ICES contains policies and procedures for its' data handling practices, and every three years these policies are reviewed and approved by the Office of the Information Privacy Commissioner [17]. At ICES, a set of data standardization rules are applied to all datasets, data cleaning is conducted, the quality of data is routinely assessed and documented using the ICES' Data Quality Framework for five different dimensions (Accuracy, Internal validity, External validity, Interpretability, and Relevance), and information about the data (e.g. data quality reports, how the data is collected) are held in an internal website on the ICES Intranet. In this study the Canadian Institute for Health Information Discharge Abstract Database (CIHI-DAD) and Same Day Surgery (SDS) database were utilized to obtain data on

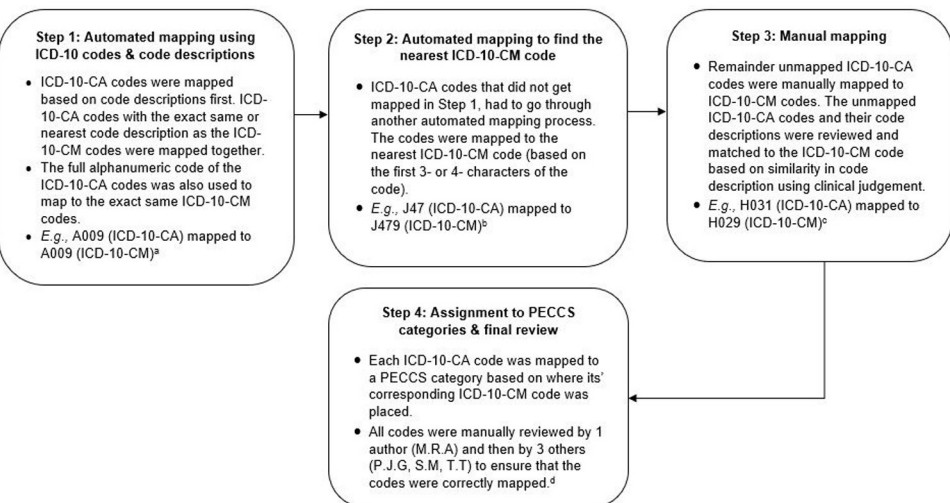

**Fig 2. Diagram outlining the step-by-step procedure used to translate PECCS from ICD-10-CM to ICD-10-CA for use in Canada.** [a] The ICD-10 code description for A009 in ICD-10-CA and ICD-10-CM is *'Cholera, unspecified'*. [b] The ICD-10-CA code description for J47 is *'Bronchiectasis'*, and the ICD-10-CM code description for J479 is *'Bronchiectasis, uncomplicated'*. [c] The ICD-10-CA code description for H031 is *'Involvement of eyelid in other infectious diseases classified elsewhere'*, and the ICD-10-CM code description for H029 is *'Unspecified disorder of eyelid'*. [d] During the review, we aimed to retain as much of the PECCS categories for ICD-10-CA from the original PECCS categories from the ICD-10-CM. However, if needed, we also modified some categories by merging original PECCS categories that overlapped or created new categories to ensure that the codes fitted within the category.

the inpatient and same day surgery hospital encounters including the admission date, age at admission, and the main responsible diagnosis recorded for the encounter using ICD-10-CA codes. This study was approved by the research ethics board at the Hospital for Sick Children, and followed the Strengthening the Reporting of Observational Studies in Epidemiology (STROBE) reporting guideline. Since deidentified administrative data were used, patient consent was waived.

### Study population and statistical analysis

The study population included children less than 18 years of age with an inpatient or same day surgery hospital encounter between April 1, 2014 to March 31, 2019. Hospital encounters among children with missing or invalid dates (i.e. birth, death, discharge), encounters with a negative value for age at admission, encounters with a discharge date after March 31, 2019, encounters among non-Ontario residents, encounters with zero cost data, and encounters with the most responsible diagnosis code for normal newborn births, residual codes with no procedures performed during the encounter or external cause codes were excluded.

To illustrate the utility of PECCS-CA, we applied PECCS-CA on the hospital encounter data and identified the ten most prevalent conditions and their volume of encounters. An overview of the steps used to develop PECCS-CA from PECCS including its' application on administrative data from Ontario can be found in Fig 1. Data were analyzed using SAS Enterprise Guide version 7.1 (SAS Institute, Inc).

### Results

The conversion of PECCS from ICD-10-CM to ICD-10-CA resulted in mapping 16,992 ICD-10-CA codes into 781 clinically distinctive condition categories. Of the 781 categories, 777 (99.5%) were from the original PECCS, 3 (0.4%) were created from merging original categories, and 1 (0.1%) was newly created. Mapping discrepancies were observed for some ICD-10-CA codes that did not get mapped to their corresponding ICD-10-CM codes, using the automated function. Examples of mapping issues that were observed are presented in Table 1. These mapping discrepancies occurred, because the alphanumeric codes or their descriptions varied between the two ICD-10 systems. An appropriate ICD-10-CM code along with their corresponding PECCS-CA category had to be manually assigned (Table 1). Another issue observed was that some ICD-10 codes present in ICD-10-CA were not available in ICD-10-CM, thus, these codes had to be manually mapped to the next most appropriate ICD-10-CM code based on the clinical condition (Table 1).

This study included 911,732 hospital encounters in children in Ontario. The PECCS-CA was applied to the hospital encounter data, which classified the encounters into 727 PECCS-CA condition categories. Fig 3 presents the ten most prevalent conditions in children with hospital encounters, including important pediatric conditions such as bronchiolitis and neonatal hyperbilirubinemia. The most prevalent condition was low birth weight (n = 54,100 encounters).

The PECCS-CA also contained several non-specific condition categories (e.g. those that start with "Other"), which were derived from the original PECCS. Although, the aim of PECCS-CA was to have mutually exclusive clinically relevant categories, it was necessary to have some non-specific categories to group ICD-10 diagnosis codes that were heterogeneous and were not appropriate to be placed in any of the other clinically relevant categories. Table 2 presents an example of three non-specific pediatric categories from PECCS-CA and the top 10 most responsible ICD-10-CA diagnoses that led to hospital encounters for each category in children in Ontario.

**Table 1. Examples of mapping issues and final decisions made in mapping ICD-10-CA codes to ICD-10-CM codes and their corresponding PECCS-CA categories.**

| ICD-10-CA Code | ICD-10-CA Code Description | ICD-10-CM Code[a] (Matched) | ICD-10-CM Code Description[a] (Matched) | PECCS-CA Category Based on Match[a] | ICD-10-CM Code[b] (Assigned) | ICD-10-CM Code Description[b] (Assigned) | PECCS-CA Category based on Assigned[b] |
|---|---|---|---|---|---|---|---|
| **Issue #1: Mapping discrepancies observed** | | | | | | | |
| E100 | Type 1 diabetes mellitus with coma | E1011 | Type 1 diabetes mellitus with ketoacidosis with coma | Diabetic ketoacidosis | E10641 | Type 1 diabetes mellitus with hypoglycemia with coma | Type 1 diabetes mellitus with complications |
| E11319 | Type 2 diabetes mellitus with preproliferative retinopathy, level of control unspecified | E11319 | Type 2 diabetes with unspecified diabetic retinopathy without macular edema | Type 2 diabetes mellitus with complications | E1139 | Type 2 diabetes with other diabetic ophthalmic complication | Type 2 diabetes mellitus with complications |
| F55 | Abuse of non-dependence-producing substances | F550 | Abuse of antacids | Substance-related disorders | F558 | Abuse of other non-psychoactive substances | Substance-related disorders |
| K670 | Chlamydial peritonitis | K67 | Disorders of peritoneum in infectious diseases classified elsewhere | Peritonitis and intestinal abscess | A7481 | Chlamydial peritonitis | Peritonitis and intestinal abscess |
| K671 | Gonococcal peritonitis | K67 | Disorders of peritoneum in infectious diseases classified elsewhere | Peritonitis and intestinal abscess | A5485 | Gonococcal peritonitis | Sexually transmitted infections (not HIV or hepatitis) |
| **Issue #2: Codes that were present in ICD-10-CA, but were not available in ICD-10-CM** | | | | | | | |
| F000 | Dementia in Alzheimer's disease with early onset | N/A | N/A | N/A | G300 | Alzheimer's disease with early onset | Delirium dementia and amnestic and other cognitive disorders |
| K020 | Caries limited to enamel | N/A | N/A | N/A | K029 | Dental caries, unspecified | Dental caries |
| K021 | Caries of dentine | | | | | | |
| K022 | Caries of cementum | | | | | | |
| K024 | Odontoclasia | | | | | | |
| K025 | Caries with pulp exposure | | | | | | |
| K028 | Other dental caries | | | | | | |

Abbreviations: ICD-10-CA, International Statistical Classification of Diseases and Related Health Problems, Tenth Revision Canada; ICD-10-CM, International Statistical Classification of Diseases, Tenth Revision, Clinical Modification; PECCS-CA, Pediatric Clinical Classification System for use in Canada; N/A, Not available.

[a] ICD-10-CM codes and code descriptions that were first matched to the ICD-10-CA code along with the corresponding PECCS-CA category during the automated mapping process.

[b] For Issue#1, the ICD-10-CM codes and code descriptions were proposed and assigned during the manual review stage, as the first matched ICD-10-CM code did not map adequately with the ICD-10-CA code. For Issue #2, there were no ICD-10-CM codes that were mapped using automated functions to the ICD-10-CA codes, thus, an ICD-10-CM code was proposed and manually assigned prior to the manual review stage. The PECCS-CA category was assigned to the ICD-10-CA code based on which PECCS category the corresponding ICD-10-CM code was placed.

## Discussion and conclusion

This study converted PECCS from ICD-10-CM to ICD-10-CA to be used in Canada (PECCS-CA) using a detailed step wise process which included automation and manual review. The conversion process resulted in categorizing 16,992 ICD-10-CA codes into 781 mutually exclusive, clinically important categories including important pediatric conditions in inpatient settings and treatments (e.g. chemotherapy). The PECCS-CA was then applied to health administrative data of pediatric hospital encounters in Ontario, the most populous province of Canada, to evaluate its' face validity and identify the most prevalent conditions in children. The PECCS-CA can be utilized to examine trends in healthcare services use and cost,

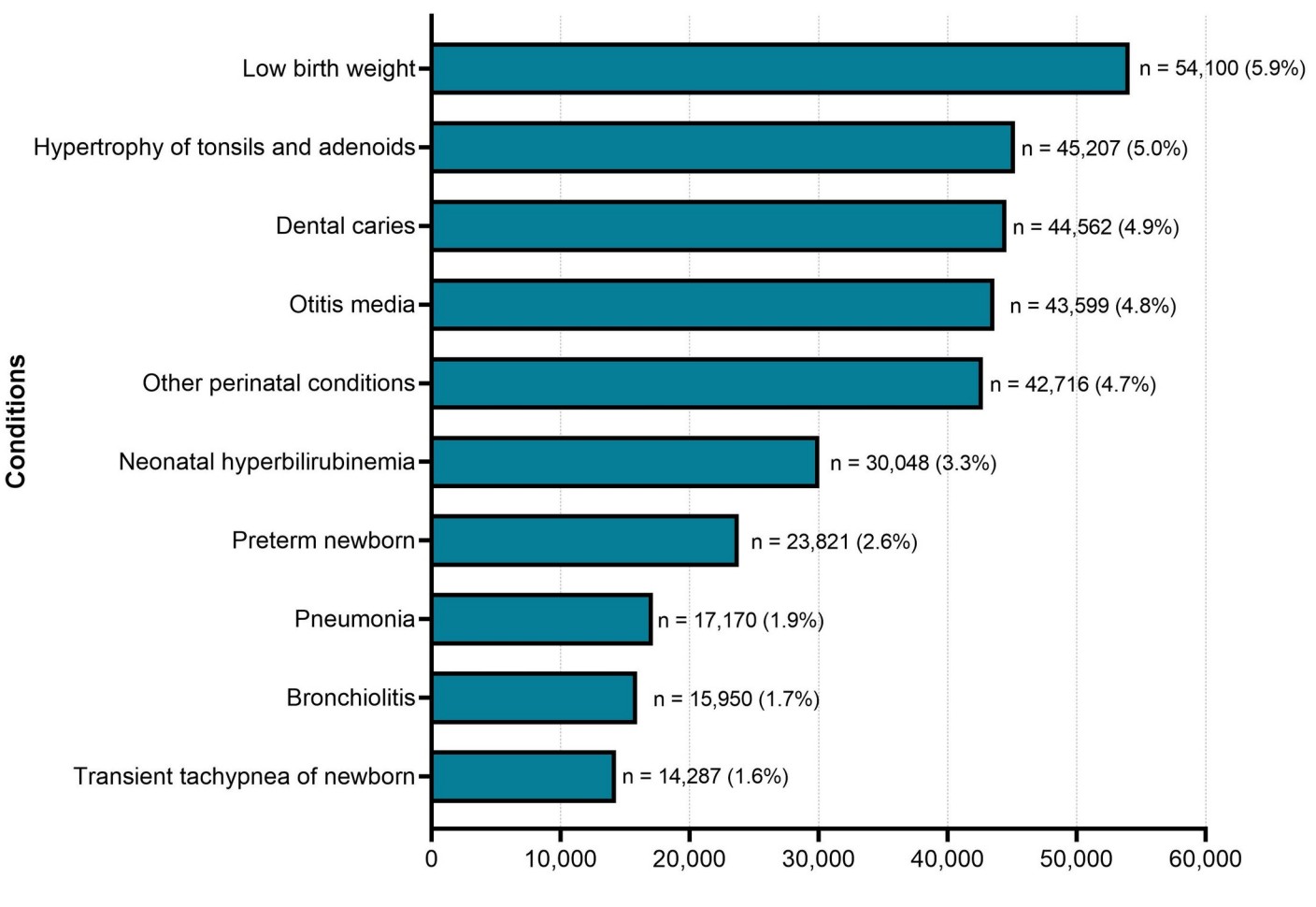

**Fig 3. Top 10 most prevalent conditions identified by applying PECCS-CA to administrative data of pediatric hospital encounters in Ontario, 2014–2019.** This figure focuses on the 10 most prevalent conditions in children with hospital encounters in Ontario during 2014–2019. Data for pediatric hospital encounters (inpatient discharges, same day surgery) were obtained from the Canadian Institute for Health Information Discharge Abstract Database (CIHI-DAD) and Same Day Surgery (SDS) database. (n = total number of hospital encounters for the condition; % = percentage of hospital encounters for the condition from all hospital encounters. The percentages displayed do not equal to 100%, because only the top 10 most prevalent conditions are presented).

rank healthcare use by conditions for research prioritization, and conduct outcomes research in pediatrics.

The PECCS-CA is the first classification system specific to pediatrics that includes important pediatric conditions (e.g. bronchiolitis, neonatal hyperbilirubinemia) found in children admitted in hospitals. This classification system has also been recently used to identify conditions that should be prioritized for research in hospitalized children based on the prevalence, cost, and variation in cost of pediatric hospitalizations in Ontario, Canada [18].

Although other grouping methodologies that categorize ICD-10-CA codes into clinical groups exist, their limitations makes them difficult to use in hospital pediatric research [3–5]. One classification system mapped ICD-10-CA codes into 130 mutually exclusive chronic health condition categories [3], however, its focus on chronic conditions limits its use in pediatric research in inpatient settings where children can be diagnosed with acute infectious diseases (e.g. bronchiolitis) or have injuries (e.g. fractures) [18]. Other existing grouping methodologies for ICD-10-CA includes the ICD-10-CA chapters [4] and the CIHI Case Mix

**Table 2. Examples of non-specific pediatric conditions in PECCS-CA and the 10 most responsible ICD-10-CA diagnosis codes that led to hospital encounters for the conditions in children in Ontario, 2014–2019.**

| Condition | Total Number of Hospital Encounters | ICD-10-CA Diagnosis Codes[a] | ICD-10-CA Diagnosis Code Descriptions[a] | No. (%) of Encounters[b] |
|---|---|---|---|---|
| Other skin disorders | 2,315 | L905 | Scar conditions and fibrosis of skin | 629 (27.2) |
| | | L720 | Epidermal cyst | 580 (25.1) |
| | | L989 | Disorder of skin and subcutaneous tissue, unspecified | 113 (4.9) |
| | | L721 | Trichilemmal cyst | 106 (4.6) |
| | | L918 | Other hypertrophic disorders of skin | 99 (4.3) |
| | | L729 | Follicular cyst of skin and subcutaneous tissue, unspecified | 92 (4.0) |
| | | L929 | Granulomatous disorder of skin and subcutaneous tissue, unspecified | 72 (3.1) |
| | | R610 | Localized hyperhidrosis | 65 (2.8) |
| | | L988 | Other specified disorders of skin and subcutaneous tissue | 60 (2.6) |
| | | L732 | Hidradenitis suppurativa | 42 (1.8) |
| | | *Other ICD-10-CA codes* | *Other ICD-10-CA code descriptions* | 457 (19.7) |
| Other nutritional, endocrine, and metabolic disorders | 1,460 | R634 | Abnormal weight loss | 467 (32.0) |
| | | E835 | Disorders of calcium metabolism | 116 (7.9) |
| | | R638 | Other symptoms and signs concerning food and fluid intake | 108 (7.4) |
| | | E713 | Disorders of fatty-acid metabolism | 70 (4.8) |
| | | R629 | Lack of expected normal physiological development, unspecified | 56 (3.8) |
| | | R630 | Anorexia | 55 (3.8) |
| | | E711 | Other disorders of branched-chain amino-acid metabolism | 49 (3.4) |
| | | E710 | Maple-syrup-urine disease | 47 (3.2) |
| | | E806 | Other disorders of bilirubin metabolism | 46 (3.2) |
| | | E740 | Glycogen storage disease | 45 (3.1) |
| | | *Other ICD-10-CA codes* | *Other ICD-10-CA code descriptions* | 401 (27.5) |
| Other nervous system disorders | 1,447 | G510 | Bell's palsy | 85 (5.9) |
| | | G934 | Encephalopathy, unspecified | 79 (5.5) |
| | | R132 | Esophageal dysphagia | 73 (5.0) |
| | | Z462 | Fitting and adjustment of other devices related to nervous system and special senses | 61 (4.2) |
| | | R251 | Tremor, unspecified | 59 (4.1) |
| | | R2688 | Other and unspecified abnormalities of gait and mobility | 58 (4.0) |
| | | G373 | Acute transverse myelitis in demyelinating disease of central nervous system | 53 (3.7) |
| | | G939 | Disorder of brain, unspecified | 49 (3.4) |
| | | G08 | Intracranial and intraspinal phlebitis and thrombophlebitis | 48 (3.3) |
| | | G540 | Brachial plexus disorders | 48 (3.3) |
| | | *Other ICD-10-CA codes* | *Other ICD-10-CA code descriptions* | 834 (57.6) |

Abbreviations: ICD-10-CA, International Statistical Classification of Diseases and Related Health Problems, Tenth Revision Canada.

[a] Top 10 most responsible ICD-10-CA diagnosis codes and corresponding code descriptions that led to hospital encounters for each non-specifc pediatric condition.

The remainder of the ICD-10-CA codes that led to hospital encounters for the condition category are grouped under '*Other ICD-10-CA codes*'.

[b] Indicates the number and percentage of hospital encounters due to each ICD-10-CA diagnosis code within the corresponding condition.

Group (CMG) [5, 6]. Although the ICD-10-CA chapters only contains 23 broad category chapters, it further breaks down into more detailed subcategories [4]. Regardless, diagnosis codes for some important pediatric conditions are categorized together, limiting its' utility to differentiate between important conditions. For instance, transient tachypnea of newborn, a distinct condition category identified in Keren *et al.*'s ICD-9-CM pediatric diagnosis code grouper [11], and the tenth most prevalent condition found in our study using PECCS-CA, was grouped under the ICD-10-CA Chapter XVI 'certain conditions originating in the perinatal period', and further categorized under the diagnosis codes for 'respiratory distress of newborn' [4, 19]. Therefore, this condition may have not been identified as prevalent if the ICD-10-CA chapters were used instead. As for the CIHI CMGs patient classification system, it uses both diagnosis and intervention codes from hospital records to classify inpatients into clinical groups [5, 6], and is not publicly available to be used. Conversely, the full-set of PECCS-CA codes is available online [20].

The CCS is a grouper used to classify the ICD-10-CM codes into clinically meaningful categories [9], and the beta version (2019.1) of the CCS was used to develop the original PECCS for the ICD-10-CM codes [7, 8]. The beta version of the CCS categorized more than 70,000 ICD-10-CM diagnosis codes into 283 clinical categories [21]. The utility of PECCS-CA was not compared to the CCS using the same hospital encounter dataset due to the differences in the ICD-10 coding systems. However, we previously compared the original PECCS's ability to detect pediatric conditions with the CCS using pediatric hospitalization data from the US [7]. The PECCS demonstrated increased specificity of detecting pediatric health conditions. For instance, 13,261 pediatric hospital encounters in the US were classified into miscellaneous mental health disorders using CCS, while the same encounters were classified into the following when PECCS was used: miscellaneous mental health disorders (5,357 encounters), anorexia nervosa (4,709 encounters), conversion disorder (2,979 encounters), and bulimia nervosa (216 encounters) [7]. If the CCS was able to be applied to our current dataset, important pediatric conditions including bronchiolitis, neonatal hyperbilirubinemia, and transient tachypnea of newborn would not have been detected. This further demonstrates the importance of PECCS-CA for pediatric health services research in Canada.

There were a number of non-specific condition categories found in PECCS-CA as these categories came directly from the original PECCS for ICD-10-CM codes [7, 8]. Some examples of these categories include: other skin disorders; other nutritional, endocrine, and metabolic disorders; and other nervous system disorders. In the original PECCS, we minimized the number of non-specfic categories as much as possible as the ICD-10-CM codes within the category were heterogenous and the category itself did not have much clinical value. In addition, similar to the process done by HCUP [9], the number of ICD-10 codes within each non-specific category was minimized as much as possible by segregating out codes that can be rather placed in other clinically relevant categories. Nevertheless, non-specific conditions are also present in other existing classification systems [3, 5, 9].

There are some important limitations of PECCS-CA. First, it does not identify if the condition is acute or chronic. However, it is still effective to be used with different data sources and at different pediatric settings [7]. Second, it contains some non-specific conditions (e.g. those that start with "Other"). Last, PECCS-CA cannot be applied to datasets in countries outside of Canada due to the different versions of ICD-10 system present across countries. Nevertheless, it can be modified to be used with other country-specific versions of the ICD-10 system.

In conclusion, this study aimed to present a translated version of PECCS from ICD-10-CM to ICD-10-CA to be used in Canada. PECCS-CA is an open-source classification system that categorizes ICD-10-CA diagnosis codes into 781 clinically meaningful categories to identify pediatric specific conditions including treatments. It can be used by researchers from different

pediatric fields and for different purposes which includes understanding the trends in healthcare services use and cost, rank the healthcare use by conditions, and to conduct patient outcomes research in pediatrics. Future works can include translating the PECCS for use with other country-specific versions of the ICD-10 classficiation system to be used internationally.

## Supporting information

**S1 Data. PECCS-CA for ICD-10-CA codes (update—February 09, 2022).**
(XLSX)

## Acknowledgments

This study was supported by ICES, which is funded by an annual grant from the Ontario Ministry of Health (MOH) and the Ministry of Long-Term Care (MLTC). ICES is an independent, non-profit research institute whose legal status under Ontario's health information privacy law allows it to collect and analyze health care and demographic data, without consent, for health system evaluation and improvement.

The opinions, results and conclusions reported in this paper are those of the authors and are independent from the funding sources. No endorsement by ICES or the Ontario MOH is intended or should be inferred. The funding sources had no role in the study design, data collection and analysis, decision to publish, or preparation of the manuscript. Parts of this material are based on data and information compiled and provided by the Canadian Institute for Health Information (CIHI). However, the analyses, conclusions, opinions, and statements expressed herein are those of the author, and not necessarily those of CIHI.

**Canadian Paediatric Inpatient Research Network (PIRN):** Patricia C Parkin MD, Ann Bayliss MD, Ronik Kanani MD, Sean Murray MD, Catherine Pound MD MSc, Mahmoud Sakran MD, Anupam Sehgal MD, Sepi Taheri MD, Gita Wahi MD PhD, Peter J Gill MD DPhil, and Sanjay Mahant MD MSc. The lead author for PIRN is Peter J Gill, and his contact email address is peter.gill@sickkids.ca.

## Author Contributions

**Conceptualization:** Peter J. Gill, Thaksha Thavam, Mohammed Rashidul Anwar, Sanjay Mahant.

**Data curation:** Jingqin Zhu.

**Formal analysis:** Mohammed Rashidul Anwar, Jingqin Zhu.

**Funding acquisition:** Peter J. Gill, Teresa To, Sanjay Mahant.

**Investigation:** Peter J. Gill, Sanjay Mahant.

**Methodology:** Peter J. Gill, Thaksha Thavam, Mohammed Rashidul Anwar, Jingqin Zhu, Teresa To, Sanjay Mahant.

**Project administration:** Peter J. Gill, Thaksha Thavam, Mohammed Rashidul Anwar, Sanjay Mahant.

**Resources:** Peter J. Gill, Sanjay Mahant.

**Software:** Mohammed Rashidul Anwar, Jingqin Zhu.

**Supervision:** Peter J. Gill, Teresa To, Sanjay Mahant.

**Validation:** Peter J. Gill, Sanjay Mahant.

**Visualization:** Peter J. Gill, Thaksha Thavam, Sanjay Mahant.

**Writing – original draft:** Peter J. Gill, Thaksha Thavam, Sanjay Mahant.

**Writing – review & editing:** Peter J. Gill, Thaksha Thavam, Mohammed Rashidul Anwar, Jingqin Zhu, Teresa To, Sanjay Mahant.

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
